# Integrative Approaches for the Diagnosis and Management of Erosive Oral Lichen Planus

**DOI:** 10.3390/diagnostics14070692

**Published:** 2024-03-26

**Authors:** Cristina Popa, Ana Maria Sciuca, Bianca-Andreea Onofrei, Stefan Toader, Oana Mihaela Condurache Hritcu, Cristina Boțoc Colac, Elena Porumb Andrese, Daciana Elena Brănișteanu, Mihaela Paula Toader

**Affiliations:** 1Discipline of Oral Medicine, Oral Dermatology, Grigore T. Popa University of Medicine and Pharmacy, 16 Universitatii Street, 700115 Iasi, Romania; dr.cristinapopa@gmail.com (C.P.); ana.filioreanu@umfiasi.ro (A.M.S.); andreea.turcanu@yahoo.com (B.-A.O.); oana.condurache-hritcu@umfiasi.ro (O.M.C.H.); mihaela.toader@umfiasi.ro (M.P.T.); 2Discipline of Physiopathology, Grigore T. Popa University of Medicine and Pharmacy, 16 Universitatii Street, 700115 Iasi, Romania; 3Dermatology Clinic, University Clinical Railways Hospital, 1 Garabet Ibraileanu Street, 700115 Iasi, Romania; 4Discipline of Dermatology, Grigore T. Popa University of Medicine and Pharmacy, 16 Universitatii Street, 700115 Iasi, Romania; elena.andrese1@umfiasi.ro (E.P.A.); daciana.branisteanu@umfiasi.ro (D.E.B.)

**Keywords:** erosive oral lichen planus, physiopathology, diagnosis, dermoscopy, topical corticosteroids, systemic immunosuppressants, biologics, JAK inhibitors, patient management

## Abstract

Erosive oral lichen planus (EOLP) represents a significant challenge in dental and medical management due to its chronic inflammatory nature, painful symptoms, and impact on quality of life. This study aims to evaluate the current diagnostic approach with novel non-invasive techniques, such as dermoscopy, and also the landscape of treatment options for EOLP, focusing on its efficacy, safety, and the challenges that it present in clinical practice. Through a comprehensive literature review, we explored the use of topical corticosteroids, systemic immunosuppressants, biologics, and Janus kinase (JAK) inhibitors in treating EOLP, alongside examining patient compliance, psychological impacts, and the risk of adverse effects and recurrence. Our findings reveal that while topical corticosteroids are the cornerstone of EOLP treatment, offering symptomatic relief, their long-term use is limited by side effects and tolerance development. Systemic therapies and biologics provide alternatives for refractory cases but necessitate careful adverse effect monitoring. JAK inhibitors show promise as an innovative treatment avenue but require more evidence on long-term safety and efficacy. This study highlights the necessity of personalized treatment approaches due to the variable disease course and response to treatment, underscoring the importance of a multidisciplinary strategy in managing EOLP. The complexity of EOLP treatment, compounded by its psychological and quality of life impacts, demands ongoing research into targeted therapies, the establishment of standardized treatment protocols, and the development of effective outcome measures to improve patient care and treatment outcomes.

## 1. Introduction

Oral lichen planus (OLP) is a chronic mucosal disease driven by T-cell-mediated inflammatory processes, mainly affecting the stratified squamous epithelia. Its multifaced clinical manifestations often present clinicians with difficulties in diagnosis and treatment, primarily stemming from its persistent and recurring nature [1].

Etiopathogenesis is incompletely understood, but research shows that epithelial keratinocytes serve as the focal point for cytotoxic T-cell reactions in the context of lichen planus (LP). The infiltrating lymphocytes in LP predominantly consist of cytotoxic CD8+ and CD45RO+ memory T-cells expressing alpha and beta T-cell receptors (TCRs). These T-cells possess the capacity to recognize a specific LP antigen on the affected keratinocytes that is associated with major histocompatibility complex class I (MHC class I). The array of implicated antigen types encompasses self-reactive peptides and exogenous antigens, such as modified proteins, medications, contact allergens, and infectious agents. Initial antigen recognition may involve antigen-presenting cells [1,2]. The activation of cytotoxic cells results in the release of cytokines, including interleukins 2, 4, and 10, gamma interferon, and alpha tumor necrosis factor (TNF). These cytokines exhibit a combination of both pro- and anti-inflammatory profiles characteristic of Th1 and Th2 responses. Consequently, cytotoxic cell activation can lead to oligo- or, occasionally, polyclonal proliferation. The clinical behavior of LP is influenced by the delicate balance between lymphocyte activation and regulatory mechanisms. In response to cytokines, basal layer keratinocytes upregulate the expression of intercellular adhesion molecules (ICAMs-1) and vascular cell adhesion molecules (VCAMs-1), facilitating local immune responses [2,3,4]. In contrast, the role of CD4+ lymphocytes in LP pathogenesis remains less elucidated. Notably, serum concentrations of TNF were significantly elevated in LP patients compared to healthy individuals, with the severity of lesions paralleling this increase. A speculative correlation exists between angiogenesis and the expression of vascular endothelial growth factor (VEGF) across various clinical forms of OLP, with the erosive variants demonstrating the strongest association. Erosive OLP is more likely to exhibit heightened levels of metalloproteinases MMP-1 and 3, which participate in basement membrane alteration and induction of the apoptotic process, compared to reticular forms [5,6]. The pronounced proliferative activity and chromosomal instability observed in erosive OLP can be linked to an overexpression of Ki67 and p53 proteins, particularly within the context of the significant increase in cell turnover observed in lichen lesions. These factors are implicated in the potential transformation of these lesions into squamous cell carcinoma. Furthermore, autoimmune mechanisms may play a role in the disease, as indicated by higher serum levels of autoantibodies directed against anti-desmoglein 1 and 3 in erosive OLP compared to reticular LP or individuals with recurrent ulcers. The erosive nature of LP lesions may also contribute to this autoimmune aspect of the disease [7,8]. Although the main cause of erosive lichen planus (ELP) is unknown, the onset of the disease is linked to several risk factors, the most common being medication and dental procedures that cause a lichenoid reaction. Different environmental risk factors have been described as potential triggers of ELP, such as various microorganisms, lifestyle, allergens, and nutritional status. Often a genetically susceptible patient is thought to be at risk of developing ELP once they experience different systemic disorders, anxiety, and sleep-related disturbance [8,9].

The oral presentation of ELP manifests as distinctive large ulcerations that are slightly depressed and have irregular contours. These ulcers are typically painful and exhibit a symmetrical distribution. The Koebner phenomenon is evident at the mucosal level, whereby traumatic events exacerbate the erosive lesions. Patients commonly report sensations of burning, metallic taste, irritation, or bleeding during oral hygiene practices, such as brushing. The entire oral cavity can be affected, with gingival involvement representing nearly 40% of the affected sites. Gingival participation is accompanied by non-specific desquamative gingivitis, characterized by erythema and edema of both marginal and attached gingiva, predominantly on the vestibular side. Application of slight pressure to the affected gingival area often results in epithelial desquamation, the formation of hemorrhagic vesicles, and the development of superficial ulcerated regions [2,10]. The additional presentations encompass atrophic and bullous configurations, regarded as distinct iterations of erosive lichen planus. Atrophic OLP manifests as diffusely distributed erythematous patches encircled by delicate white streaks, thereby inducing symptomatic discomfort in affected individuals. Conversely, the bullous variant predominantly manifests on the buccal mucosa and the lateral margins of the tongue, subsequently rupturing to give rise to burning and erosive lesions [10]. Genital lesions associated with ELP are primarily situated in the posterior vestibule, with potential extension to the labia minora. These genital lesions are typified by erythema and erosions, often encircled by characteristic white lace-like borders. ELP-induced genital lesions lead to significant alterations in the normal anatomical structure of the affected area [9,10].

The diagnosis of erosive oral lichen planus (EOLP) is primarily clinical and relies on the appearance of the lesions and associated symptomatology and their progression over time. While a microscopic examination can provide supporting evidence for the diagnosis, its principal purpose is the identification of potential malignant transformation of suspicious lesions. When conducting microscopic examinations, it is crucial to obtain samples from the periphery of EOLP lesions for optimal diagnostic accuracy [11].

## 2. Materials and Methods

Study Motivation: Treating patients with erosive lichen planus (ELP) presents significant challenges due to the chronic, relapsing nature of the disease and its profound impact on patients’ quality of life. One of the primary difficulties in managing ELP stems from its unpredictable response to treatment, as therapies that are effective in some patients may yield minimal or no benefit in others. The mainstay treatment options, including topical corticosteroids and systemic immunosuppressants, often carry the risk of adverse effects, particularly with long-term use, necessitating a careful balance between managing the disease and minimizing potential harm to the patient. Additionally, the psychological burden of ELP, characterized by chronic pain, functional impairment, and concerns about the aesthetic implications of oral lesions, can complicate treatment adherence and effectiveness. The potential for malignant transformation in chronic lesions further complicates the management strategy, requiring ongoing surveillance and multidisciplinary collaboration. This complexity underscores the need for individualized treatment plans, incorporating both pharmacologic interventions and supportive care, to address the multifaceted challenges faced by patients with erosive lichen planus.

Search Strategy: A search of the PubMed and Medline databases was performed according to the Preferred Reporting Items for Systematic Reviews guidelines for articles regarding the treatment options for erosive lichen planus published both in English and French. Using keywords such as “oral erosive lichen planus”, “oral erosive lichen planus diagnosis”, “erosive lichen planus”, “mucosal lichen planus dermoscopy”, “oral erosive lichen planus treatment”, “topical corticosteroids”, and “systemic corticosteroids”, a total of 140 articles were selected, which included case reports, review articles, clinical studies, and meta-analyses. For each publication included, we recorded the author(s), study year, study design, localization, therapies used, number of treated patients, treatment outcomes, and adverse effects.

Study Selection: Two independent reviewers screened the title and abstract of all identified articles to determine their eligibility for inclusion. Full-text articles were then reviewed to confirm eligibility. Inclusion criteria were (1) studies investigating the actual and potential therapies for ELP, (2) studies published both in English and French, (3) studies including human subjects, and (4) studies reporting original data.

Data Extraction: Data were extracted from the included studies by two independent reviewers using a standardized form. The following data were extracted: (1) study design, (2) sample size, (3) patient characteristics (e.g., age, gender, previous therapies), and (4) outcomes.

Data Synthesis: Due to the heterogeneity of study designs and outcomes, we conducted a narrative synthesis of the data. We grouped studies based on the type of treatment and discussed the findings within each group.

## 3. Results

### 3.1. Diagnostic Approach

In 1978, the World Health Organization (WHO) established the initial diagnostic criteria for OLP, encompassing clinical and histopathological parameters. However, these criteria failed to address the differentiation of epithelial dysplasia from OLP, leading to substantial inconsistencies among observers. In 2003, van der Meji and van der Waal proposed revisions to these criteria. Their first step was to distinguish between OLP and oral lichenoid lesions (OLL) criteria. Clinically, OLP diagnosis primarily hinges on the presence of bilateral white reticulum, rather than the presence of plaque, atrophic, erosive, or bullous lesions. From a histopathological perspective, the absence of dysplasia, such as atypical cell morphology, is explicitly mentioned as a requirement for diagnosing OLP. This was an effort to exclude lichenoid dysplasia. The authors recommended the use of terms like “histopathologically compatible with” or “clinically compatible with” when either microscopic or clinical features were less apparent. Elements such as varied inflammatory infiltration, deeper submucosal infiltration extending beyond the superficial connective tissue, and perivascular infiltration were highlighted as indicative of OLL rather than OLP. Achieving complete alignment between the clinical and histopathological criteria proposed by van der Meij and van der Waal, often referred to as the modified WHO diagnostic criteria, is imperative for making a diagnosis of OLP [11]. The diagnosis of erosive oral lichen planus (EOLP) is primarily clinical, requiring a thorough examination and a detailed patient history to distinguish it from similar mucosal conditions. Key diagnostic features include the presence of painful, erythematous lesions with ulcerations, often accompanied by Wickham’s striae on the mucosa [1].

Dermatoscopy, when applied to the oral mucosa (mucoscopy), significantly enhances the diagnostic approach to erosive oral lichen planus (EOLP). This non-invasive technique provides a magnified view of EOLP’s distinctive morphological characteristics, facilitating differentiation from similar mucosal lesions. Research, including that by Sonthalia et al. (2018) [11] and Rouai et al. (2021) [12], underscores its value in improving diagnostic precision, monitoring disease evolution, and minimizing the reliance on invasive diagnostic procedures. The integration of dermatoscopy into the clinical evaluation of EOLP represents a pivotal improvement in oral medicine, promising more accurate diagnoses and informed therapeutic decisions. Thus, the most common dermatoscopic feature of mucosal lichen planus is represented by Wickham’s striae, found in 91% of the cases, with a reticular, radial, annular, or globular pattern [12]. Other white lesions of OLP present dermoscopically in a homogenous pattern, as a veil-like structure, with a bluish or grey hue. The vascularity is regularly distributed, with linear blood vessels most commonly found, but dotted, looped, and radial vessels may also be present [12]. The erosions are seen in dermoscopy as shiny, bright red homogenous lesions (Figure 1). Other dermoscopic features of OLP may include pigmented lesions, scales, and blunted lingual papillae [12].

A biopsy of the lesion is critical for confirming the diagnosis. Histopathological examination of hematoxylin–eosin-stained tissue samples reveals a band-like lymphocytic infiltrate just below the epithelial layer, hyperkeratosis, and sometimes, Civatte bodies (apoptotic keratinocytes). Direct immunofluorescence can further aid in diagnosis by showing fibrinogen deposits in a shaggy pattern along the basement membrane.

While there are no specific biomarkers for EOLP, the differential diagnosis should rule out conditions such as pemphigus vulgaris, mucous membrane pemphigoid, and other forms of lichen planus affecting different body sites [5].

The evolution of EOLP is typically chronic with acute exacerbations and rarely self-limiting, posing significant therapeutic challenges [13,14]. Malignant transformation, estimated to be between 0.3% and 5%, is the primary progressive risk of OLP, despite its controversies, particularly in the oldest, chronic, atrophic, or erosive forms [13,14,15].

### 3.2. Therapeutic Options

According to the literature, the treatment options for erosive lichen planus are various, but in the absence of a consensus guideline, the approach to the disease can be difficult and, often, non-curable. A wide range of treatment regimens have been described, including the use of topical therapies, such as topical steroids, calcineurin inhibitors, phototherapy, and systemic therapies, such as systemic corticosteroids, methotrexate, and cyclosporine [16]. Table 1 summarizes the topical and systemic therapies for EOLP found in the literature.

#### 3.2.1. Topical Therapies

##### Topical Steroids

Due to their ability to target superficial inflammation in the epidermis, upper dermis, and dermo-epidermal junction, topical steroids are considered to be the first-line treatment option for ELP. Due to its minimal systemic absorption, topical therapy is frequently prescribed for cases with limited lesions, thereby avoiding the adverse effects of systemic corticotherapy. However, a recent study revealed that topical steroids administered for more than six months can suppress the HPA axis despite the absence of clinical symptoms suggestive of adrenal insufficiency [17,18].

One of the main topical steroids used is clobetasol, which has excellent anti-inflammatory properties that target edema, fibrin deposits, vasodilation, and phagocytic activity [19]. As a result, topical clobetasol is a first-line treatment option for many clinicians in the treatment of ELP [20,21]. Traditionally, clobetasol can be administered in various forms, including aqueous solutions and orafix ointments, with variable mucosal adhesion properties [22]. Several studies have attempted to enhance the mucosal adhesion of topical clobetasol by modifying its formulation, showing statistically significant and clinically meaningful effectiveness in both objective and subjective enhancement while maintaining a favorable safety record [23]. By combining 4% hydroxyethyl cellulose gel with an equivalent amount of clobetasol to obtain a concentration of 0.05%, Arduino et al. were able to improve the oral mucosal delivery of the drug; therefore, the clobetasol group had a better clinical improvement than the placebo group [24].

Triamcinolone acetonide intralesional administration can be an effective treatment option for ELP with minimal systemic adverse effects [25,26]. The literature cites minimal local side effects like candidiasis, irritations, and hyperpigmentations [26,27]. However, due to the drug’s moderate potency and brief half-life, the disease is likely to recur according to Xia et al., Kuo et al., and Lee et al. [28,29,30].

Another topical steroid used is betamethasone, which presents itself in various formulations available on the market, including mouthwash and paste [31]. The anti-inflammatory, analgesic, and immunosuppressive properties of betamethasone allow it to be used as a treatment for ELP according to Liu et al. and supported by Carbone et al. [32,33]. Liu et al. administered a betamethasone compound containing two forms, betamethasone dipropionate and betamethasone disodium phosphate, intralesionally as an alternative to the classical form. This compound provided a rapid peak of the effect due to the first form of the drug and a slower absorption to prolong the effect due to the second form [32].

##### Topical Calcineurin Inhibitors

Calcineurin inhibitors are second-line options to be considered in cases where topical steroids have failed to alleviate the signs and symptoms of ELP without causing significant side effects. A total of 0.1% tacrolimus ointment and 1% pimecrolimus cream are the most commonly used topical calcineurin inhibitors (TCIs) [34].

Long-term use of TCIs is associated with transient burning sensations, taste alterations, frequent relapses, and a higher likelihood of oral squamous cell carcinoma [35].

A double-blind controlled trial conducted by Arduino et al. on 30 patients concluded that after 2 months of treatment, the patients treated with 1% pimecrolimus showed more long-term stability of the treatment compared with those treated with 0.1% tacrolimus. Complete resolution of the symptoms was noted for four patients. Side effects were also noted for a small group of patients, including a transient burning sensation and sialorrhea for the tacrolimus group, and xerostomia, gastroesophageal reflux, and reactivation of a herpes labialis infection for the pimecrolimus group [36].

Despite the fact that the majority of studies showed consistent improvement of the ELP lesion treated with 0.1% tacrolimus, according to Kiyani et al.’s study, 0.1% tacrolimus is not an option for all the patients as a long-term treatment for ELP. Their follow-up of the patients showed suboptimal response for half of the patients, with relapses of the disease [37].

One of the main concerns of the topical administration of the treatment is the adhesion to the oral mucosa. Ibrahim et al. reported statistically insignificantly superior results when using tacrolimus as an oral patch carrier system compared to tacrolimus gel and topical steroids. However, the compliance of the patients was significantly higher for the oral patch carrier group and the initial results were better, with the patients showing a faster clinical response [38].

The main treatment options TCIs are compared to are topical steroids; Sun et al. reported comparable beneficial effects between the two groups [39]. According to Pakfetrat et al., ELP lesions treated with triamcinolone acetonide 1% in orabase and pimecrolimus 1% showed comparable improvement. These results are consistent with prior research performed by Azizi et al. and Gorouhi et al. [40,41,42]. While several studies showed similar efficacy between tacrolimus and topical steroids. Chamani et al. noted a slightly higher improvement of the tacrolimus group compared to the clobetasol group, suggesting that TCIs may be a viable option as a first-line treatment option for ELP [43,44,45,46].

##### Nd:YAG Laser

The Nd:YAG laser, operating at a wavelength of 1064 nm, exhibits a unique property wherein it is preferentially absorbed by hemoglobin while displaying limited absorption by water. This characteristic imparts a notable penetration depth of up to 5–6 mm to the laser. The 1064 nm wavelength is less absorbed by the chromophore hemoglobin compared to the 530–600 nm wavelength, thereby enabling deeper tissue penetration compared to conventional laser systems. Furthermore, the Nd:YAG laser induces tissue coagulation during its transit through the tissue matrix [47]. Khater et al. conducted an evaluation of the therapeutic efficacy of Nd:YAG laser in the context of treating erosive lichen planus lesions, revealing a reduction in discomfort levels and an improvement in clinical scores. The parameters employed in this study encompassed an energy range of 80–120 mJ, a frequency of 6–15 Hz, a non-contact handpiece, a spot size diameter of 0.9 mm, pulse durations ranging from 100 ls (very short pulse) to 300 ls (short pulse), fluences between 12.6 and 18.9 J/cm^2^, and an air/water mist ratio of 6/5. The utilization of this specific wavelength engenders an efficient and expeditious healing process with minimal discomfort experienced both during and after the intervention [48]. Hu et al. and Wang et al. used the Nd:YAG laser combined with oral total glucosides of paeony to treat ELP, the results demonstrating the potential to enhance the effectiveness of treating EOLP, offering a safe and efficient treatment option suitable for broad clinical utilization [49,50].

##### Photodynamic Therapy and Low-Level Laser Therapy

In light of the adverse effects associated with systemic corticotherapy, continuous exploration of therapeutic modalities with minimal or negligible side effects is imperative. Photobiomodulation, also known as low-level laser therapy (LLLT), exerts favorable effects, including a reduction in local inflammation, a promotion of wound healing, tissue regeneration, pain alleviation, and an inhibition of the fibrosis process [51]. Photodynamic therapy (PDT) emerges as a relatively recent treatment approach applicable to various medical conditions. PDT involves the amalgamation of three integral components: a source of oxygen, a light source operating at a specific wavelength (typically 660 nm), and a photosensitizer, such as methylene blue. PDT presents several advantages, notably its minimally invasive nature, absence of residual lesions, and independence from anesthesia. Encouragingly, PDT has demonstrated promise in the treatment of oral lesions [52,53]. Mirza et al. conducted a comparative assessment of the efficacy of photodynamic therapy, LLLT, and topical steroids for OELP lesions. The results indicated similar improvements, with a slight advantage favoring PDT over LLLT. Conversely, individuals treated with topical steroids experienced a substantial reduction in pain scores. Due to their anti-inflammatory properties, which mitigate prostaglandin E2, interleukin 1 beta, tumor necrosis factor-alpha, oxidative stress, and tissue edema, both PDT and LLLT represent viable avenues for prospective research, as corroborated by the studies of He et al. and Mutafchieva et al. Two additional investigations by Rakesh et al. and Nammour et al. substantiate the findings of Mirza et al. [54,55,56,57,58]. As elucidated by Mahdavi et al., the beneficial effects of phototherapy on ELP lesions are contingent on parameters such as wavelength, power, intensity, exposure duration, session frequency, and treatment sequence [59].

##### Local UVB

In recent years, phototherapy has emerged as an adjunctive therapeutic strategy for managing OLP, particularly in its refractory forms. The current constraints of this approach include the potential carcinogenicity associated with PUVA therapy, as well as the elevated costs associated with extracorporeal photochemotherapy and PDT [60,61]. Kassem et al. have presented UVB phototherapy as a promising therapeutic modality for erosive lichen planus lesions, especially those that remain unresponsive to conventional treatments. Clinical improvements have been observed when minimal erythemal concentrations of UVB are administered three times weekly [62]. Notably, UVB phototherapy does not confer an enhanced photosensitivity risk, in contrast to PUVA therapy, which necessitates the systemic administration of psoralen, thereby inducing associated adverse effects, and PDT, which entails the administration of aminolevulinic acid [63].

##### Curcumin

Curcumin, derived from the turmeric plant, possesses notable therapeutic attributes, encompassing anti-inflammatory, antioxidant, and anticarcinogenic properties. Additionally, it exerts immunomodulatory effects through interactions with macrophages and natural killer cells [64,65]. Recent systematic reviews have shed light on the potential of topically administered curcumin in the context of erosive lichen planus (ELP), although it is important to note that it may not constitute a viable substitute for topical corticosteroids in the majority of patients [66,67]. However, due to the safety of curcumin, it may be considered as an adjunct therapy alongside corticosteroids for the improvement of discomfort, burning sensations, and the clinical presentation of oral lesions in patients with OLP [68]. Notably, a study conducted by Nosratzehi et al. centered on the local application of a mucoadhesive turmeric paste over a 12-week period, leading to observed enhancements in terms of lesion dimensions, pain reduction, and alleviation of burning sensations. Consequently, it was concluded that the topical utilization of curcumin can be an effective and well-tolerated approach in the treatment of ELP lesions [69]. It is worth noting that the primary side effects reported by patients include digestive discomfort, which tends to correlate with the dosage of the medication [70].

##### Ozonized Water

Ozone exhibits a range of therapeutic attributes, including immunomodulatory properties, the alleviation of pain, antioxidant capabilities, antimicrobial effects, and the promotion of expedited wound healing processes [71]. Notably, research by Veneri et al. has underscored the advantageous impact of combining ozone therapy with topical steroids compared to the use of topical steroids alone. This combined approach demonstrates superior efficacy in terms of pain management and the mitigation of ELP lesions [72]. Furthermore, the formulation of ozone in the form of ozonized water presents several merits, particularly in facilitating the administration of treatment, even in cases involving oropharyngeal lesions. It emerges as a potentially swifter, safer, and more user-friendly alternative to gaseous ozone therapy, thereby enhancing its clinical utility [73,74].

##### Cryotherapy

The advantages associated with cryotherapy in the oral cavity encompass the absence of pain in contrast to surgical procedures, minimal scarring, straightforward application, localized therapeutic action devoid of systemic adverse effects, and a notably low incidence of infections. This modality is also cost-effective and offers portable units that can be easily employed within clinical settings [75]. However, it is imperative to acknowledge certain limitations. Cryotherapy may exhibit unpredictable expansion patterns, lack precision in terms of depth control, and is subject to operator proficiency. Pain, albeit generally of minor intensity, can arise due to the cellular breakdown products affecting peripheral nerves proximal to the treatment site. Such discomfort can be effectively managed using non-narcotic analgesic agents [76]. In a study conducted by Amanat et al., a comparative analysis was performed between the favorable outcomes of a single cryotherapy session and the effects of topical applications of 0.1% tacrolimus and 0.1% NOG. Results revealed that both groups exhibited comparable pain scores and lesion severity, underscoring the potential of cryotherapy as an adjunct therapeutic approach for patients with ELP. This adjunctive utilization of cryotherapy could potentially reduce the reliance on medication. Nevertheless, it is noteworthy that the recurrence of lesions was observed, implying that long-term treatment outcomes may not align entirely with initial expectations [77].

Table 2 summarizes the topical therapies applied for EOLP, outlining the outcome and side effects for each therapeutic agent, as well as the level of evidence derived from our literature search.

#### 3.2.2. Systemic Therapies

##### Systemic Corticotherapy

Systemic corticotherapy, encompassing agents such as prednisone, prednisolone, methylprednisolone, betamethasone, and dexamethasone, represents the predominant therapeutic approach in the management of ELP. This intervention finds frequent application in cases characterized by severity, extensive lesion involvement, or acute exacerbations. Additionally, systemic corticosteroids emerge as a viable resource when topical treatments fail to yield desired outcomes. In clinical practice, a combination of oral and topical corticosteroids is frequently deployed to address ELP lesions comprehensively [78,79]. The requisite dosages are meticulously tailored, considering the lesion’s extent, the patient’s body weight, and their prior response to therapy. However, it is imperative to adopt a cautious approach when considering long-term systemic corticotherapy due to the propensity for diminished therapeutic efficacy over time. To mitigate potential rebound effects, a gradual dose reduction is mandated, typically spanning a period of 4 to 6 weeks upon achieving remissions [80]. Nonetheless, it is important to acknowledge that sustained reliance on systemic corticosteroids is rarely deemed a sustainable treatment long-term therapeutic strategy for ELP management. Side effects associated with this option encompass the development of iatrogenic Cushing syndrome and the suppression of the hypothalamic–pituitary–adrenal axis. Additionally, susceptibility to infections, ulcerative or peptic conditions (attributable to prostaglandin inhibition), glaucoma, cataracts, dermatological manifestations (including acneiform eruptions and increased hair growth), and neurological disturbances (such as behavioral changes, anxiety, insomnia, and mood alterations) constitute noteworthy side effects resulting from prolonged and systemic glucocorticoid usage [81,82,83].

##### Azathioprine

Azathioprine, classified as an immunosuppressive agent, functions as a purine metabolism antagonist, impeding the synthesis of deoxyribonucleic acid (DNA), ribonucleic acid (RNA), and proteins [84]. While it traditionally served as a primary therapeutic option for autoimmune disorders, including lupus, rheumatoid arthritis, psoriatic arthritis, and inflammatory bowel disease, its prominence has waned in favor of mycophenolate mofetil (MMF), extensive investigations indicating that azathioprine exhibits lower efficacy when compared to MMF [84,85]. Azathioprine assumes a role in the treatment of erosive and generalized OLP as a steroid-sparing therapeutic alternative, particularly in severe instances of ELP resistant to topical interventions. Typical initial dosing commences at 1.0 mg/kg/day (50 to 100 mg), with gradual increments of 0.5 mg/kg/day over several weeks, if warranted, up to a maximum of 2.0 mg/kg/day. Vigilant monitoring of complete blood counts is advisable during the initial eight weeks of treatment. If no discernible improvement materializes after three months, discontinuation of azathioprine is recommended [86]. Azathioprine’s foremost adverse effect is acute myelosuppression, stemming from the accumulation of 6-thioguanine, particularly pronounced in individuals with deficient or absent TPMT enzyme activity. Moreover, prolonged usage of this agent carries an elevated risk of infections and cancer, alongside gastrointestinal toxicity [87,88].

##### Cyclosporine

Historically employed in topical therapy through the rinse and spit technique, cyclosporine exhibited variable results owing to its susceptibility to inactivation via cytochrome P450-dependent biotransformation within the mucosa, resulting in subtherapeutic systemic levels [89]. Gupta et al. explored the utilization of oral cyclosporine in treating EOLP and reported complete remission of lesions after a three-month treatment period. This suggests a potential role for cyclosporine as a third-line therapy for erosive lichen planus. In a separate study in 2017, Kilic et al. employed oral cyclosporine to treat a rare case of plantar erosive lichen planus, initiating treatment at 3 mg/kg per day and progressively tapering the dosage. Although lacking follow-up details, the authors observed significant lesion improvement during the course of treatment [90,91]. Oral cyclosporine has found its use in a refractory to most topical and systemic therapy cases of erosive genital lichen planus (EGLP) [92].

##### Mycophenolate Mofetil

Mycophenolate mofetil (MMF), an immunosuppressive agent, inhibits leukocyte recruitment and adhesion to endothelial cells, exerting a cytostatic effect on lymphocytes. Multiple investigations have substantiated the efficacy of systemic MMF in ameliorating recalcitrant and refractory ELP lesions [93,94]. Deen et al. documented a case where a patient with EGLP received 500 mg of MMF for three months, resulting in significant improvements in erosions and pruritus, corroborating previous findings by Wee et al. and Frieling et al. [95,96,97]. Notably, common adverse effects of MMF encompass gastrointestinal issues, lymphopenia, anemia, urinary symptoms, susceptibility to infections, and elevated liver enzymes [93].

##### Rituximab

Rituximab, a chimeric murine–human monoclonal CD20 antibody, licensed for treating vasculitis, B-cell lymphomas, and rheumatoid arthritis, operates through multiple mechanisms, including signal pathway interruption, complement-mediated lysis, antibody-dependent cytotoxicity, and apoptosis induction [98]. Heelan et al. documented successful treatment of ELP with rituximab, following the protocol for rheumatoid arthritis. Lagerstedt et al. reported a case where a patient with EGLP, unresponsive to prior treatments (topical and systemic corticosteroids, hydroxychloroquine, and methotrexate), exhibited a favorable response to rituximab. The mechanisms behind lesion remission may involve peripheral blood B-cell depletion, the downregulation of T-CD8 cell-activating cytokines, or autoantibody production. However, Tetu et al.’s case series involving rituximab treatment for ELP suggested varying degrees of effectiveness, with older participants displaying less favorable responses [99,100,101].

##### Methotrexate

Methotrexate (MTX), an acid analogue of folic acid, inhibits purine and pyrimidine synthesis, with dose-dependent activity: inhibiting folate-dependent enzymes at high doses to inhibiting IL-1 activity at lower doses [102]. Several studies propose low-dose oral methotrexate as an option for severe and resistant forms of EOLP, while others indicate similar efficacy at moderate doses. Lajevardi et al. conducted a study in 2015, treating 18 patients with EOLP resistant to topical or systemic therapy with methotrexate (15 mg/week for 12 weeks). Significant pain relief, a partial response in at least 83.3% of patients, and a notable decrease in the Thongprasom scale were observed. Cline et al. conducted a study on 27 patients with ELPV who were unresponsive to other treatments, with 56% of patients demonstrating clinical improvement [103,104]. Management of EGLP lacks strong evidence-based data for systemic treatment options, although Cline et al. suggest MTX as a possible option to be taken into consideration with more than 50% of their treated patients maintaining the therapeutic response [104]. Adverse effects of long-term methotrexate therapy may encompass elevated liver enzymes, hepatic fibrosis, hematological changes, mucocutaneous disorders, and the need for ongoing liver and kidney function monitoring [102].

##### Systemic Tacrolimus

As a calcineurin inhibitor, tacrolimus impedes calcium-dependent processes, including interleukin-2 gene transcription, nitric oxide synthase activation, cell degranulation, and apoptosis. While topical tacrolimus is established in ELP treatment, systemic tacrolimus is infrequently considered [105]. Yeo et al. pioneered a study employing oral tacrolimus (0.5–0.15 mg/kg/day for at least six weeks) in two ELP patients, with dose-dependent adverse effects. Both patients exhibited marked lesion improvement. Chen et al. conducted a similar study in 2017 involving three patients, yielding comparable results at half the cited dose [106,107].

##### Hydroxychloroquine

Hydroxychloroquine, by influencing lysosomal activity, autophagy, membrane stability, signaling pathways, and transcriptional activity, reduces toll-like receptor signaling and cytokine production [108]. Studies suggest hydroxychloroquine may be viable for EOLP forms unresponsive to topical therapies. Morrison et al. conducted a study on 30 EOLP patients, reporting at least mild improvements and no significant side effects. Vermeer et al. treated 15 ELPV patients with hydroxychloroquine (200–800 mg daily), resulting in significant lesion improvement. Yeshurun et al. conducted a study on 21 previously treated EOLP patients who received hydroxychloroquine sulfate (400 mg/day for 1 to 36 months), with 81% of patients benefiting from the treatment. In another study by Raj et al. in 2021, 45 patients received six months of treatment with 200 mg of hydroxychloroquine, resulting in notable improvement [109,110,111,112].

##### Apremilast

Apremilast, categorized as a phosphodiesterase 4 inhibitor, exerts inhibitory effects on the production of multiple pro-inflammatory cytokines, including TNF alpha, IFN gamma, IL-2, IL-5, IL-8, and IL-12 [113]. A study conducted by Paul et al. observed the effects of oral apremilast at a dose of 20 mg administered twice daily to ten patients diagnosed histopathologically with ELP. One patient exhibited a notable improvement in oral lesions by the study’s conclusion. Given the demonstrated safety profile of apremilast in prior investigations for psoriasis, it is posited as a promising therapeutic option for ELP [114]. AbuHilal et al. presented a case study in which a patient suffering from EOLP and desquamative gingivitis received apremilast at a dose of 30 mg twice daily for ten days, followed by tapering to 30 mg once daily for twelve weeks. The authors concluded that apremilast showcased potential viability in EOLP treatment, given the considerable reduction in lesions following the treatment regimen. Moreover, Hafner et al. reported a case of ELP coupled with stenotic esophagitis, which exhibited complete remission of oral erosive lesions after four weeks of apremilast therapy. These findings align harmoniously with those of Bettencourt et al., further substantiating the potential utility of apremilast [115,116,117]. It is crucial to note that the primary adverse effects associated with apremilast pertain to susceptibility to infections and gastrointestinal complications [118].

##### Eiconol

Eiconol, a derivative derived from marine fish oil, comprises a complex of unsaturated fatty acids endowed with lipid-lowering properties. Scientific investigations have indicated that a daily intake of 6 g of fish oil can enhance the clinical presentation of patients afflicted with EOLP [119,120].

##### Selenium

Selenium, an essential component of a well-balanced diet, serves a multitude of pivotal biological functions, encompassing the support of normal thyroid gland function, the inhibition of tumor growth, and the exertion of antioxidant effects [121]. A study conducted by Belal et al. unveiled that the management of EOLP patients exhibited improvement through the administration of a cocktail consisting of Vitamin A, C, and E, and selenium (100 µg) once daily, in conjunction with a combination of dexamethasone and itraconazole [122]. According to a study conducted by Qataya et al., selenium has the potential to serve as an adjunctive therapy to the primary treatment for ELP, whether administered topically or systemically. It is noteworthy that all three groups in the study exhibited amelioration of the lesions; however, it is particularly noteworthy that the two groups receiving selenium as a component of their treatment demonstrated a sustained enhancement in lesion condition at the end of the study [123].

##### Etanercept

Etanercept, operating as a TNF receptor antagonist and impeding immune and inflammatory cell production, primarily garners application in rheumatoid arthritis and severe psoriatic arthritis. Nonetheless, investigations have ascertained its utility in treating EOLP lesions that manifest resistance to conventional treatments. Noteworthy drawbacks encompass the high cost associated with its administration and the proclivity for recidivism upon cessation of the drug [124]. In a review conducted by Didona et al., there have been cases cited in the past treated with anti-TNF-alpha agents, the most representative one being a case report realized by Yarom et al. in 2007. The case study involved a patient afflicted with severe EOLP, unresponsive to systemic topical agents, retinoids, and azathioprine but responsive to etanercept at a subcutaneous dosage of 25 mg twice weekly. Substantial mucosal improvement and alleviation of pain commenced in the fourth week but were curtailed in the tenth week due to cost constraints, leading to a relapse after a subsequent two weeks [86]. Notable adverse effects cited include opportunistic infections and demyelination [125].

##### Basiliximab

Basiliximab, a monoclonal antibody designed to inhibit IL-2R clonal expansion, has been employed successfully in the management of severe orogenital erosive LP. However, it is observed that this intervention may present as a short-term solution, with disease relapse occurring approximately four weeks post-treatment [126]. Principal adverse effects include the formation of neutralizing antibodies and cytokine release syndrome [127].

##### Efalizumab

Efalizumab, a humanized monoclonal antibody impeding T lymphocyte function, has exhibited promise in the context of EOLP treatment [128]. Deng et al. reviewed a series of past clinical trials that proposed efalizumab as a potential therapeutic option for ELP. Cheng et al. reported the significant improvement of a patient with EOLP resistant to prednisone and topical tacrolimus following a five-week regimen of efalizumab at a dosage of 0.7 mg/kg. Similar positive outcomes were witnessed in a patient with generalized LP after 12 weeks of efalizumab treatment who was also administered at 0.7 mg/kg. However, Hefernan et al.’s unicentric, open-label pilot study involving four patients underscored the effectiveness of efalizumab (0.7 mg/kg in week 0, followed by 1.0 mg/kg through week 12) in EOLP treatment. Nevertheless, two patients experienced severe adverse effects, such as urticaria, staphylococcal abscess of a hip prosthesis, and drug-induced subacute cutaneous lupus [129,130,131]. Pertinent concerns regarding the safety profile of efalizumab, including the risk of progressive multifocal leukoencephalopathy, require careful consideration [132].

Table 3 summarizes the systemic therapeutic options for EOLP, with or without association with a topical agent, their outcome, side effects, and level of evidence as found in the literature search.

#### 3.2.3. Future Therapeutic Options

Current treatments for erosive lichen planus (ELP) are limited, with off-label options being explored for severe cases resistant to standard therapies. Early and thorough management is crucial due to the significant impact on patient quality of life and the disease’s progression, but topical therapies alone often fall short of effective control [133].

The JAK-STAT signaling pathway is crucial in oral lichen planus (OLP) development, facilitating cytokine signaling within cells. It involves specific proteins and enzymes that, upon activation, regulate gene transcription critical for immune and hematologic functions. Genetic mutations in this pathway can lead to various diseases, highlighting its significance. In OLP, dysfunction of this pathway contributes to the disease’s inflammatory nature. The understanding of this pathway’s role in such diseases has spurred the development of JAK inhibitors, promising new treatments for dermatological conditions, although they are not yet FDA-approved for these indications [134].

Topical JAK inhibitors are formulated to selectively suppress the activities of certain JAK kinases, thus regulating the body’s inflammatory mechanisms. These inhibitors achieve this by curtailing phosphorylation and the subsequent activation of STAT proteins, which, in turn, lower the expression of genes that trigger inflammation [135]. Their utility in managing OLP and other skin-related conditions stems from their capability to attenuate local inflammation and immune responses upon direct application to affected areas. This localized approach helps in concentrating the therapeutic effects while potentially lessening the likelihood of widespread side effects. Yet, the overall safety profiles of these medications are subject to ongoing scrutiny, with a particular focus on their impact on immune system modulation [136]. Exploring Tofacitinib has highlighted its potential in addressing difficult cases of lichen planus, including those histologically akin to OLP, suggesting its viability as a targeted therapy for particularly stubborn or severe instances of OLP [137]. Research into Baricitinib has yielded positive outcomes in the context of OLP, indicating its effectiveness. Its role in adjusting the immune response, a critical factor in OLP’s onset, positions it as a promising option for treatment. While detailed analyses of Ruxolitinib’s role in OLP are not as abundant as those for Tofacitinib and Baricitinib, its proven effectiveness across a spectrum of lichen planus suggests that it could also be relevant for OLP management. Consideration of Upadacitinib in the treatment of lichen planus variants, some of which closely resemble OLP, points to its potential efficacy in addressing OLP specifically. The potential of JAK inhibitors in treating oral lichen planus (OLP) offers new avenues for patients dealing with this persistent and distressing condition. Continuous research is crucial, with a focus on thoroughly understanding the long-term safety and effectiveness of these treatments, especially considering their effects on the immune system. Developing consistent methods for measuring outcomes and treatment guidelines will be key to better integrating JAK inhibitors into clinical practice [138]. So far, the evidence suggests that JAK inhibitors could significantly change how we manage OLP, offering hope for more effective treatments. The healthcare community is keenly awaiting further research to expand the options available for people with OLP [139]. The ongoing exploration of novel targeted therapies may expand into unconventional domains. Considering the involvement of the mast cells in the inflammatory cascade and subsequent cytokines release post-degranulation, it is plausible to posit that mast cell stabilizers could potentially mitigate these effects, thereby attenuating the inflammatory response. Consequently, mast cell stabilizers could represent a prospective avenue for adjunctive therapy research in EOLP, although further investigations are imperative to substantiate this hypothesis [140].

## 4. Discussion and Conclusions

The management of erosive oral lichen planus (EOLP) remains a challenging endeavor for clinicians due to the disease’s chronic nature, the potential for significant morbidity, and the risk of malignant transformation. Treatment strategies primarily aim at reducing symptoms, minimizing mucosal damage, and preventing complications, yet no standard protocol guarantees remission. The cornerstone of therapy involves topical corticosteroids, which, despite their effectiveness in reducing inflammation and pain, may not be suitable for all patients, especially considering the potential for adverse effects and the development of resistance with long-term use.

Advancements in therapeutic options have introduced systemic agents, including immunosuppressants, biologics, and most recently, JAK inhibitors, offering hope for refractory cases. However, these systemic treatments come with their own set of challenges, including the need for careful monitoring of side effects and the consideration of individual patient factors, such as comorbid conditions and concomitant medications. The efficacy of these agents varies, underscoring the importance of personalized treatment approaches based on the severity of the disease, patient preferences, and response to previous therapies.

Furthermore, the psychological impact of EOLP cannot be understated, as chronic pain and aesthetic concerns significantly affect patients’ quality of life, potentially complicating treatment adherence and effectiveness. This highlights the need for a multidisciplinary approach to management, incorporating dental care, pain management, psychological support, and regular surveillance for potential malignant transformation.

Despite the array of available treatments, the unpredictable nature of EOLP and its variable response to therapy illustrate the complexities of managing this condition. Ongoing research into the pathogenesis of EOLP and the development of targeted therapies is crucial. Additionally, the establishment of standardized treatment protocols and outcome measures will enhance our ability to evaluate the efficacy of novel treatments. Ultimately, improving the quality of life for patients with EOLP requires not only advancements in medical treatment but also a comprehensive approach that addresses the physical, emotional, and social impacts of this debilitating disease.

## Figures and Tables

**Figure 1 diagnostics-14-00692-f001:**
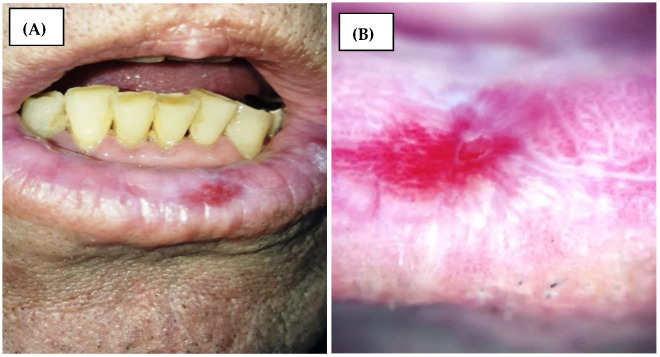
(**A**) Clinical presentation of erosive oral lichen planus on the lower lip; (**B**) dermatoscopic view of erosive oral lichen planus on the lower lip showing a bright red erosion, grey-white veil, linear and radial Wickham striae, and globular blood vessels with a regular distribution.

**Table 1 diagnostics-14-00692-t001:** Topical and systemic therapies for EOLP.

TOPICAL THERAPIES	SYSTEMIC THERAPIES
Corticosteroids - Clobetasol - Triamcinolone acetonide - Betamethasone Calcineurin inhibitors - Tacrolimus - Pimecrolimus Phototherapy - Nd:YAG laser - Photodynamic therapy and low-lever laser therapy - Local UVB phototherapy Curcumin Ozonized water Cryotherapy	Corticosteroids - Prednisone - Methylprednisolone - Betamethasone - Dexamethasone Azathioprine Cyclosporine Mycophenolate mofetil Rituximab Systemic tacrolimus Methotrexate Hydroxychloroquine Apremilast Eiconol Selenium Etanercept Basiliximab Efalizumab

**Table 2 diagnostics-14-00692-t002:** Topical therapies for EOLP and their level of evidence: outcome, recurrence rate, and side effects.

AUTHOR	TOPICAL TREATMENT	OUTCOME (%) CASES	RR	F/U	SIDE EFFECTS	CA	LEVEL OF EVIDENCE
Brennan et al. [22]	Clobetasol	PR	N/A	4 weeks	Salivary hypersecretion Nausea, diarrhea Application site pain Application site hemorrhage	N/A	Level 2
Arduino et al. [23]	Clobetasol	PR	37%	24 weeks	Gastroesophageal reflux Severe skin erythematous reaction Mild increase in blood sugar	N/A	Level 2
Kuo et al. [29]	Triamcinolone acetonide Dexamethasone	CR (90%)	100%	N/A	Oral candidiasis	N/A	Level 4
Wei Zhao et al. [25]	Triamcinolone acetonide	CR (95.8%)	37.5%	12 weeks	Mild dry mouth	N/A	Level 2
Chuanxia Liu et al. [31]	Betamethasone	CR (93.1%)	14.8%	12 weeks	Slight burning sensation in the throat	N/A	Level 2
Arduino et al. [35]	Tacrolimus 0.05% Pimecrolimus 0.5%	CR (33%) CR (35%)	73.3% 26.6%	24 weeks	Mucosal burning sensation Transient sialorrhea Gastroesophageal reflux Xerostomia Recurrence of herpes labialis	N/A	Level 2
Kyiani et al. [36]	Tacrolimus 0.1%	CR (50%)	50%	48 weeks	Pain Intolerance to spicy food	N/A	Level 4
Ibrahim et al. [37]	Tacrolimus patch Tacrolimus gel Triamcinolone acetonide+ miconazole	PR PR PR	N/A	12 weeks	Mild burning sensation	N/A	Level 2
Pakfetrat et al. [39]	Pimecrolimus cream 1% Triamcinolone acetonide paste	CR (71.4%) CR (87.5%)	N/A	N/A	None	N/A	Level 2
Khater et al. [47]	Nd:YAG laser	CR (4.16)	N/A	N/A	N/A	N/A	Level 6
Mirza et al. [53]	PDT LLLT Topical steroids	PR	19.6% 25.9% 0.5%	N/A	None	N/A	Level 2
Mahdavi et al. [58]	LLLT	PR	0%	12 weeks	None	N/A	Level 6
Mutafchieva et al. [55]	LLLT	CR (8.33%)	N/A	N/A	N/A	N/A	Level 2
Rakesh et al. [56]	LLLT	PR	N/A	4 years	N/A	N/A	Level 2
Nammour et al. [57]	PBM	PR	87.5%	48 weeks	N/A	N/A	Level 2
Khaitan et al. [65]	Curcumin	CR (31.25%)	N/A	12 weeks	N/A	N/A	Level 3
Kia et al. [67]	Curcumin	CR (36%)	N/A	N/A	Burning sensation Itching Mild swelling Xerostomia	N/A	Level 2
Nosratzehi et al. [68]	Curcumin	PR	N/A	12 weeks	None	N/A	Level 4
Veneri et al. [71]	Betamethasone and ozonized water	PR	34.6%	N/A	Candidiasis	N/A	Level 2
Amanat et al. [76]	Cryotherapy	PR	73.3%	6 weeks	Minor swelling Pain augmentation	N/A	Level 3

RR = rate of recurrence; F/U = follow up; CA = cancer; PR = partial remission; CR = complete remission; N/A = not available.

**Table 3 diagnostics-14-00692-t003:** Systemic therapies for EOLP and their level of evidence, outcome, recurrence rate, and side effects.

AUTHOR	SYSTEMIC TREATMENT	TOPICAL TREATMENT	OUTCOME (%) CASES	RR	F/U	SIDE EFFECTS	CA	LEVEL OF EVIDENCE
Gupta et al. [89]	Cyclosporine	None	CR (100%)	N/A	12 weeks	None	N/A	Level 6
Kilic et al. [90]	Cyclosporine	None	PR	N/A	20 weeks	None	N/A	Level 6
Deen et al. [94]	Mycophenolate mofetil	None	CR (100%)	N/A	24 weeks	None	N/A	Level 5
Heelan et al. [98]	Rituximab+ Prednisone	Clobetasol	CR (100%)	50%	70 weeks	None	Breast carcinoma with spinal metastasis	Level 6
Lagerstedt et al. [99]	Rituximab	Corticosteroids	PR	N/A	64 weeks	N/A	N/A	Level 6
Tetu et al. [100]	Rituximab	None	PR	100%	N/A	N/A	N/A	Level 6
Lajevardi et al. [102]	Methotrexate	None	PR	N/A	96 weeks	Elevated liver enzymes Nausea Epigastric pain Pytiriazis rosea-like skin eruption	N/A	Level 2
Cline et al. [103]	Methotrexate	Clobetasol Tacrolimus	CR (19%)	7%	10 years	Fatigue and gastro-intestinal distress	Oral scuamocelular carcinoma Thyroid cancer	Level 4
Yeo et al. [105]	Tacrolimus	None	PR	N/A	6 weeks	N/A	N/A	Level 6
Chen et al. [106]	Tacrolimus		CR (33%)	N/A	N/A	Fatigue	N/A	Level 6
Morrison et al. [108]	Hydroxychloroquine	N/A	PR	N/A	108 weeks	N/A	N/A	Level 4
Vermeer et al. [109]	Hydroxychloroquine	-Topical steroids -HCA 2.5% estradiol Vaginal cream -Topical tacrolimus 0.1% -Intralesional steroids	PR	20%	96 weeks	Hearing loss Nausea Infection	N/A	Level 4
Yeshurun et al. [110]	Hydroxychloroquine	None	CR (24%)	N/A	144 weeks	Elevation of serum creatinine Visual field defects Hyper- pigmentation	N/A	Level 3
Raj et al. [111]	Hydroxychloroquine	N/A	PR	0%	52 weeks	N/A	N/A	Level 2
Paul et al. [113]	Apremilast	None	CR (20%)	N/A	4 weeks	Nausea	N/A	Level 3
AbuHilal et al. [114]	Apremilast	None	PR	N/A	N/A	Nausea	N/A	Level 6
Hafner et al. [115]	Apremilast	None	CR (100%)	N/A	N/A	N/A	N/A	Level 6
Bettencourt et al. [116]	Apremilast	None	PR	N/A	24 weeks	Nausea Diarrhea	N/A	Level 6
Belal et al. [121]	Selenium	Topical steroids Antifungal agent	PR	N/A	6 weeks	N/A	N/A	Level 2
Qataya et al. [122]	Selenium	None	PR	N/A	12 weeks	N/A	N/A	Level 2

RR = rate of recurrence; F/U = follow up; CA = cancer; PR = partial remission; CR = complete remission; N/A = not available.

## Data Availability

Data are available in the medical databases, according to references.

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
