# Peer review of "Integrative Approaches for the Diagnosis and Management of Erosive Oral Lichen Planus"

_diagnostics, 2024, doi:10.3390/diagnostics14070692_

Round 1

Reviewer 1 Report

Comments and Suggestions for Authors

Review of “Integrative Approaches to the Diagnosis and Management of Erosive Oral Lichen Planus”

Dear Authors,

I have read with great interest your article. You have written a review article about treatment options for erosive lichen planus. Two independent reviewers have screened the results from the literature and then summarized topical and systemic treatment options.

I have just a few remarks:

Ad 1. Line 131- should it write Items, not Itmes?

Ad 2. How did you determine the level of evidence for studies included in the article? (as cited in Table 2 and 3)  

Ad 3. I suggest you add a reference number next to the author’s surname in Table 2 and Table 3

I have no comments regarding the structure of the paper, English language use or cited references.

The manuscript is well written, has a good structure and should be interesting to the readers of Diagnostics journal. I recommend it for publication.

Reviewer 2 Report

Comments and Suggestions for Authors

given the extensive discussion of the etiopathogenesis of erosive lichen planus from a genetic point of view perhaps a bit long-winded and difficult to interpret for those without biological genetic expertise, I wonder if you have also studied the role of mast cells in the pathology? these cells as can be seen in the articles below play an important role in the patient's immune defenses and thus in the process of inflammation: 

-Tetè G, D'orto B, Ferrante L, Polizzi E, Cattoni F. Role of mast cells in oral inflammation. J Biol Regul Homeost Agents. 2021 Jul-Aug;35(4 Suppl. 1):65-70. doi: 10.23812/21-4supp1-6. PMID: 34425662.

-Kolpakov AV, Moshkova AA, Melikhova EV, Sokolova DY, Muravskaya NP, Samorodov AV, Kopaneva NO, Lukina GI, Abramova MY, Mamatsashvili VG, Parshkov VV. Diffuse Reflectance Spectroscopy of the Oral Mucosa: In Vivo Experimental Validation of the Precancerous Lesions Early Detection Possibility. Diagnostics (Basel). 2023 May 5;13(9):1633. doi: 10.3390/diagnostics13091633. PMID: 37175023; PMCID: PMC10177876.

in the materials and methods what are the exclusion criteria for found articles? did you use a precise methodology? how many were there at the beginning?

the use of lasers especially of this type in a delicate pathology like lichen erosivo seems to me excessive and also from a certain medical point of view dangerous, have you found clinical data on this?

For table 2, how did you assign levels of evidence to the items?

from the discussion is it correct to imagine that there are no standard and effective therapies for everyone from the literature review so the treatment remains unchanged?

the article as a whole does not bring any new scientific or pathogenetic insights, however, it can be a reference to quickly find references regarding the pathology. 

Comments on the Quality of English Language

English is sufficient to understand the article fluently
